# Parallel Sampling of HDPs using Sub-Cluster Splits

**Jason Chang**
CSAIL, MIT
jchang7@csail.mit.edu

**John W. Fisher III**
CSAIL, MIT
fisher@csail.mit.edu

## Abstract

We develop a sampling technique for Hierarchical Dirichlet process models. The parallel algorithm builds upon [1] by proposing large split and merge moves based on learned sub-clusters. The additional global split and merge moves drastically improve convergence in the experimental results. Furthermore, we discover that cross-validation techniques do not adequately determine convergence, and that previous sampling methods converge slower than were previously expected.

## 1  Introduction

Hierarchical Dirichlet Process (HDP) mixture models were first introduced by Teh et al. [2]. HDPs extend the Dirichlet Process (DP) to model groups of data with shared cluster statistics. Since their inception, HDPs and related models have been used in many statistical problems, including document analysis [2], object categorization [3], and as a prior for hidden Markov models [4].

The success of HDPs has garnered much interest in inference algorithms. Variational techniques [5, 6] are often used for their parallelization and speed, but lack the limiting guarantees of Markov chain Monte Carlo (MCMC) methods. Unfortunately, MCMC algorithms tend to converge slowly. In this work, we extend the recent DP Sub-Cluster algorithm [1] to HDPs to accelerate convergence by inferring "sub-clusters" in parallel and using them to propose large split moves.

Extensions to the HDP are complicated by the additional DP, which violates conjugacy assumptions used in [1]. Furthermore, split/merge moves require computing the joint model likelihood, which, prior to this work, was unknown in the common *Direct Assignment* HDP representation [2]. We discover that significant overlap in cluster distributions necessitates new *global* split/merge moves that change all clusters simultaneously. Our experiments on synthetic and real-world data validate the improved convergence of the proposed method. Additionally, our analysis of joint summary statistics suggests that other MCMC methods may converge prematurely in finite time.

## 2  Related Work

The seminal work of [2] introduced the Chinese Restaurant Franchise (CRF) and the Direct Assignment (DA) sampling algorithms for the HDP. Since then, many alternatives have been developed. Because HDP inference often extends methods from DPs, we briefly discuss relevant work on both models that focus on convergence and scalability. Current methods are summarized in Table 1.

Simple Gibbs sampling methods, such as CRF or DA, may converge slowly in complex models. Works such as [11, 12, 13, 14] address this issue in DPs with split/merge moves. Wang and Blei [7] developed the only split/merge MCMC method for HDPs by extending the Sequentially Allocated Merge-Split (SAMS) algorithm of DPs developed in [13]. Unfortunately, reported results in [7] only show a marginal improvement over Gibbs sampling. Our experiments suggest that this is likely due to properties of the specific sampler, and that a different formulation significantly improves convergence. Additionally, SAMS cannot be parallelized, and is therefore only tested on a corpus with 263K words. By designing a parallel algorithm, we test on a corpus of 100M words.

Table 1: Capabilities of MCMC Sampling Algorithms for HDPs

| | CRF [2] | DA [2] | SAMS [7] | FSD [4] | Hog-Wild [8] | Super-Cluster [9] | Proposed |
|---|---|---|---|---|---|---|---|
| Infinite Model | ✓ | ✓ | ✓ | · | ✓ | ✓ | ✓ |
| MCMC Guarantees | ✓ | ✓ | ✓ | ✓ | · | ✓ | ✓ |
| Non-Conjugate Priors | * | * | · | ✓ | · | * | ✓ |
| Parallelizable | · | · | · | ✓ | ✓ | ✓ | ✓ |
| Local Splits/Merges | · | · | ✓ | · | · | · | ✓ |
| Global Splits/Merges | · | · | · | · | · | · | ✓ |

∗ potentially possible with some adapatation of the DP Metropolis-Hastings framework of [10].

There has also been work on parallel sampling algorithms for HDPs. Fox et al. [4] generalizes the work of Ishwaran and Zarepour [15] by approximating the highest-level DP with a finite symmetric Dirichlet (FSD). Iterations of this approximation can be parallelized, but fixing the model order is undesirable since it no longer grows with the data. Furthermore, our experiments suggest that this algorithm exhibits poor convergence. Newman et al. [8] present an alternative parallel approximation related to Hog-Wild Gibbs sampling [16, 17]. Each processor independently runs a Gibbs sampler on its assigned data followed by a resynchronization step across all processors. This approximation has shown to perform well on cross-validation metrics, but loses the limiting guarantees of MCMC. Additionally, we will show that cross-validation metrics are not suitable to analyze convergence.

An exact parallel algorithm for DPs and HDPs was recently developed by Willamson et al. [9] by grouping clusters into independent super-clusters. Unfortunately, the parallelization does not scale well [18], and convergence is often impeded [1]. Regardless of exactness, all current parallel sampling algorithms exhibit poor convergence due to their local nature, while split/merge proposals are essentially ineffective and cannot be parallelized.

## 2.1 DP Sub-Clusters Algorithm

The recent DP Sub-Cluster algorithm [1] addresses these issues by combining non-ergodic Markov chains into an ergodic chain and proposing splits from learned sub-clusters. We briefly review relevant aspects of the DP Sub-Cluster algorithm here. MCMC algorithms typically satisfy two conditions: *detailed balance* and *ergodicity*. Detailed balance ensures that the target distribution is a stationary distribution of the chain, while ergodicity guarantees uniqueness of the stationary distribution. The method of [1] combines a Gibbs sampler that is *restricted* to non-empty clusters with a Metropolis-Hastings (MH) algorithm that proposes splits and merges. Since any Gibbs or MH sampler satisfies detailed balance, the true posterior distribution is guaranteed to be a stationary distribution of the chain. Furthermore, the combination of the two samplers enforces ergodicity and guarantees the convergence to the stationary distribution.

The DP Sub-Cluster algorithm also augments the model with auxiliary variables that learn a two-component mixture model for *each* cluster. These "sub-clusters" are subsequently used to propose splits that are learned over time instead of built in a single iteration like previous methods. In this paper, we extend these techniques to HDPs. As we will show, considerable work is needed to address the higher-level DP and the overlapping distributions that exist in topic modeling.

## 3 Hierarchical Dirichlet Processes

We begin with a brief review of the equivalent CRF and DA representations of the HDP [2] depicted in Figures 1a–1b. Due to the prolific use of HDPs in topic modeling, we refer to the variables with their topic modeling names. $\beta$ is the corpus-level, global topic proportions, $\theta_k$ is the parameter for topic $k$, and $x_{ji}$ is the $i$th word in document $j$. Here, the CRF and DA representations depart. In the CRF, $\tilde{\pi}_j$ is drawn from a stick-breaking process [19], and each "customer" (i.e., word) is assigned to a "table" through $t_{ji} \sim \text{Categorical}(\tilde{\pi}_j)$. The higher-level DP then assigns "dishes" (i.e., topics) to tables via $k_{jt} \sim \text{Categorical}(\beta)$. The association of customers to dishes through the tables is equivalent to assigning a word to a topic. In the CRF, multiple tables can be assigned the same dish.

The DA formulation combines these multiple instances and *directly* assigns a word to a topic with $z_{ji}$. The resulting document-specific topic proportions, $\pi_j$, aggregates multiple $\tilde{\pi}_j$ values. For

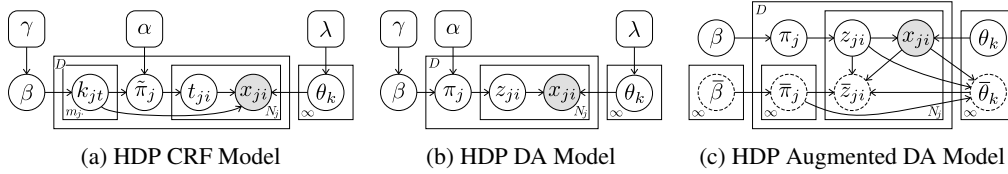

(a) HDP CRF Model     (b) HDP DA Model     (c) HDP Augmented DA Model

Figure 1: Graphical models. (c) Hyper-parameters are omitted and auxiliary variables are dotted.

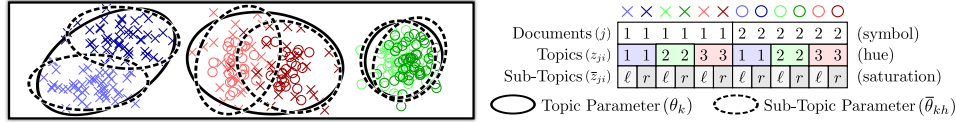

Figure 2: Visualization of augmented sample space.

reasons which will be discussed, inference in the DA formulation still relies on some aspects of the CRF. We adopt the notation of [2], where the number of tables in restaurant $j$ serving dish $k$ is denoted $m_{jk}$, and the number of customers in restaurant $j$ at table $t$ eating dish $k$ is $n_{jtk}$. Marginal counts are represented with dots, e.g., $n_{j\cdot\cdot} \triangleq \sum_{t,k} n_{jtk}$ and $m_{j\cdot} \triangleq \sum_k m_{jk}$ represent the number of customers and dishes in restaurant $j$, respectively. We refer the reader to [2] for additional details.

## 4 Restricted Parallel Sampling

We draw on the DP Sub-Cluster algorithm to combine a restricted, parallel Gibbs sampler with split/merge moves (as described in Section 2.1). The former is detailed here, and the latter is developed in Section 5. Because the restricted Gibbs sampler cannot create new topics, dimensions of the infinite vectors $\beta$, $\pi$, and $\theta$ associated with empty clusters need not be instantiated. Extending the DA sampling algorithm of [2] results in the following *restricted* posterior distributions:

$$p(\beta|m) = \text{Dir}(m_{\cdot 1}, \ldots, m_{\cdot K}, \gamma), \tag{1}$$

$$p(\pi_j|\beta, z) = \text{Dir}(\alpha\beta_1 + n_{j\cdot 1}, \ldots, \alpha\beta_K + n_{j\cdot K}, \alpha\beta_{K+1}), \tag{2}$$

$$p(\theta_k|x, z) \propto f_x(x_{\mathcal{I}_k}; \theta_k) f_\theta(\theta_k; \lambda), \tag{3}$$

$$p(z_{ji}|x, \pi_j, \theta) \propto \sum_{k=1}^{K} \pi_{jk} f_x(x_{ji}; \theta_k) \mathbb{1}[z_{ji} = k], \tag{4}$$

$$p(m_{jk}|\beta, z) = f_m(m_{jk}; \alpha\beta_k, n_{j\cdot k}) \triangleq \frac{\Gamma(\alpha\beta_k)}{\Gamma(\alpha\beta_k + n_{j\cdot k})} s(n_{j\cdot k}, m_{jk})(\alpha\beta_k)^{m_{jk}}. \tag{5}$$

Since $p(\beta|\pi)$ is not known analytically, we use the auxiliary variable, $m_{jk}$, as derived by [2, 20]. Here, $s(n, m)$ denotes unsigned Stirling numbers of the first kind. We note that $\beta$ and $\pi$ are now $(K + 1)$–length vectors partitioning the space, where the last components, $\beta_{K+1}$ and $\pi_{j(K+1)}$, aggregate the weight of all empty topics. Additionally, $\mathcal{I}_k \triangleq \{j, i; z_{ji} = k\}$ denotes the set of indices in topic $k$, and $f_x$ and $f_\theta$ denote the observation and prior distributions. We note that if $f_\theta$ is conjugate to $f_x$, Equation (3) stays in the same family of parametric distributions as $f_\theta(\theta; \lambda)$.

Equations (1–5), each of which can be sampled in parallel, fully specify the restricted Gibbs sampler. The astute reader may notice similarities with the FSD approximation used in [4]. The main differences are that the $\beta$ distribution in Equation (1) is exact, and that sampling $z$ in Equation (4) is explicitly restricted to non-empty clusters. Unlike [4], however, this sampler is guaranteed to converge to the true HDP model when combined with any split move (cf. Section 2.1).

## 5 Augmented Sub-Cluster Space for Splits and Merges

In this section we develop the augmented, sub-cluster model, which is aimed at finding a two-component mixture model containing a likely split of the data. As demonstrated in [1], these splits perform well in DPs because they improve at every iteration of the algorithm. Unfortunately, because these splits perform poorly in HDPs, we modify the formulation to propose more flexible moves.

For each topic, $k$, we fit two sub-topics, $k\ell$ and $kr$, referred to as the "left" and "right" sub-topics. Each topic is augmented with auxiliary global sub-topic proportions, $\overline{\beta}_k = \{\overline{\beta}_{k\ell}, \overline{\beta}_{kr}\}$, document-

level sub-topic proportions, $\overline{\pi}_{jk} = \{\overline{\pi}_{jk\ell}, \overline{\pi}_{jkr}\}$, and sub-topic parameters, $\overline{\theta}_k = \{\overline{\theta}_{k\ell}, \overline{\theta}_{kr}\}$. Furthermore, a sub-topic assignment, $\overline{z}_{ji} \in \{\ell, r\}$ is associated with each word, $x_{ji}$. The augmented space is summarized in Figure 1c and visualized in Figure 2. These auxiliary variables are denoted with the same symbol as their "regular-topic" counterparts to allude to their similarities. Extending the work of [1], we adopt the following auxiliary generative and marginal posterior distributions:

<div style="text-align:center">Generative Distributions        Marginal Posterior Distributions</div>

$$p(\overline{\beta}_k) = \mathrm{Dir}(\gamma, \gamma), \qquad\qquad p(\overline{\beta}_k|\bullet) = \mathrm{Dir}(\gamma + \overline{m}_{.k\ell}, \gamma + \overline{m}_{.kr}), \quad (6)$$

$$p(\overline{\pi}_{jk}|\overline{\beta}_k) = \mathrm{Dir}(\alpha\overline{\beta}_{k\ell}, \alpha\overline{\beta}_{kr}), \qquad p(\overline{\pi}_{jk}|\bullet) = \mathrm{Dir}(\alpha\overline{\beta}_{k\ell}+\overline{n}_{j\cdot k\ell}, \alpha\overline{\beta}_{kr}+\overline{n}_{j\cdot kr}), \quad (7)$$

$$p(\overline{\theta}_k|\overline{\pi}, \overline{\theta}, z, x) = \prod_{h\in\{\ell,r\}} f_\theta(\overline{\theta}_{kh}; \lambda) \prod_{j,i\in\mathcal{I}_k} Z_{ji}(\overline{\pi}, \overline{\theta}, z, x), \qquad p(\overline{\theta}_{kh}|\bullet) \propto f_x(x_{\mathcal{I}_{kh}}; \overline{\theta}_{kh}) f_\theta(\overline{\theta}_{kh}; \lambda), \quad (8)$$

$$p(\overline{z}|\overline{\pi}, \overline{\theta}, z, x) = \prod_{k=1}^{K} \prod_{j,i\in\mathcal{I}_k} \frac{\overline{\pi}_{jk\overline{z}_{ji}} f_x(x_{ji}; \overline{\theta}_{k\overline{z}_{ji}})}{Z_{ji}(\overline{\pi}, \overline{\theta}, z, x)}, \qquad p(\overline{z}_{ji}|\bullet) \propto \overline{\pi}_{jz_{ji}\overline{z}_{ji}} f_x(x_{ji}; \overline{\theta}_{z_{ji}\overline{z}_{ji}}) \quad (9)$$

$$Z_{ji}(\overline{\pi}, \overline{\theta}, z, x) \triangleq \sum_{h\in\{\ell,r\}} \overline{\pi}_{jz_{ji}h} f_x(x_{ji}; \overline{\theta}_{z_{ji}h}), \quad p(\overline{m}_{jkh}|\bullet) = f_m(\overline{m}_{jkh}; \alpha\overline{\beta}_{kh}, \overline{n}_{j\cdot kh}), \quad (10)$$

where $\bullet$ denotes all other variables. Full derivations are given in the supplement. Notice the similarity between these posterior distributions and Equations (1–5). Inference is performed by interleaving the sampling of Equations (1–5) with Equations (6–10). Furthermore, each step can be parallelized.

## 5.1 Sub-Topic Split/Merge Proposals

We adopt a Metropolis-Hastings (MH) [21] framework that proposes a split/merge from the sub-topics and either accepts or rejects it. Denoting $v \triangleq \{\beta, \pi, z, \theta\}$ and $\overline{v} \triangleq \{\overline{\beta}, \overline{\pi}, \overline{z}, \overline{\theta}\}$ as the set of regular and auxiliary variables, a sampled proposal, $\{\hat{v}, \hat{\overline{v}}\} \sim q(\hat{v}, \hat{\overline{v}}|v)$ is accepted with probability

$$\Pr[\{v, \overline{v}\} = \{\hat{v}, \hat{\overline{v}}\}] = \min\left[1, \frac{p(x,\hat{v})p(\hat{\overline{v}}|x,\hat{v})}{p(x,v)p(\overline{v}|x,v)} \cdot \frac{q(v|x,\hat{v})q(\overline{v}|x,\hat{\overline{v}},v)}{q(\hat{v}|x,v)q(\hat{\overline{v}}|x,\overline{v},\hat{v})}\right] = \min[1, H]. \quad (11)$$

$H$, is known as the *Hastings ratio*. Algorithm 1 outlines a general split/merge MH framework, where steps 1–2 propose a sample from $q(\hat{v}|x,v)q(\hat{\overline{v}}|x,v,\overline{v},\hat{v})$. Sampling the variables other than $\hat{z}$ is detailed here, after which we discuss three versions of Algorithm 1 with variants on sampling $\hat{z}$.

---
**Algorithm 1** Split-Merge Framework

---
1. Propose assignments, $\hat{z}$, global proportions, $\hat{\beta}$, document proportions, $\hat{\pi}$, and parameters, $\hat{\theta}$.
2. Defer the proposal of auxiliary variables to the restricted sampling of Equations (1–10).
3. Accept/reject the proposal with the Hastings ratio.

---

**(Step 1: $\hat{\beta}$)**: In Metropolis-Hastings, convergence typically improves as the proposal distribution is closer to the target distribution. Thus, it would be ideal to propose $\hat{\beta}$ from $p(\beta|\hat{z})$. Unfortunately, $p(\beta|z)$ cannot be expressed analytically without conditioning on the dish counts, $m_{\cdot k}$, as in Equation (1). Since the distribution of dish counts depends on $\beta$ itself, we approximate its value with

$$\tilde{m}_{jk}(z) \triangleq \arg\max_m p(m|\beta = 1/K, z) = \arg\max_m \frac{\Gamma(1/K)}{\Gamma(1/K + n_{j\cdot k})} s(n_{j\cdot k}, m)(\tfrac{1}{K})^m, \quad (12)$$

where the global topic proportions have essentially been substituted with $1/K$. We note that the dependence on $z$ is implied through the counts, $n$. We then propose global topics proportions from

$$\hat{\beta} \sim q(\hat{\beta}|\hat{z}) = p(\hat{\beta}|\tilde{m}(\hat{z})) = \mathrm{Dir}\left(\tilde{m}_{\cdot 1}(\hat{z}), \cdots, \tilde{m}_{\cdot K}(\hat{z}), \gamma\right). \quad (13)$$

We will denote $\tilde{m}_{jk} \triangleq \tilde{m}_{jk}(z)$ and $\hat{\tilde{m}}_{jk} \triangleq \tilde{m}_{jk}(\hat{z})$. We emphasize that the approximate $\hat{\tilde{m}}_{jk}$ is only used for a proposal distribution, and the resulting chain will still satisfy detailed balance.

**(Step 1: $\hat{\pi}$)**: Conditioned on $\beta$ and $z$, the distribution of $\pi$ is known to be Dirichlet. Thus, we propose $\hat{\pi} \sim p(\hat{\pi}|\hat{\beta}, \hat{z})$ by sampling directly from the true posterior distribution of Equation (2).

**(Step 1: $\hat{\theta}$)**: If $f_\theta$ is conjugate to $f_x$, we sample $\hat{\theta}$ directly from the posterior of Equation (3). If non-conjugate models, any proposal can be used while adjusting for it in the Hastings ratio.

**(Step 2)**: We use the *Deferred* MH sampler developed in [1], which sets $q(\hat{\bar{v}}|x,\hat{v}) = p(\hat{\bar{v}}|x,\hat{v})$ by deferring the sampling of auxiliary variables to the restricted sampler of Section 5. Splits and merges are then only proposed for topics where auxiliary variables have already burned-in. In practice burn-in is quite fast, and is determined by monitoring the sub-topic data likelihoods.

**(Step 3)**: Finally, the above proposals results in the following the Hastings ratio:

$$H = \frac{p(\hat{\beta},\hat{z})p(x|\hat{z})}{p(\beta,z)p(x|z)} \cdot \frac{q(z|\hat{v},\hat{\bar{v}})q(\beta|z)}{q(\hat{z}|v,\overline{v})q(\hat{\beta}|\hat{z})}. \tag{14}$$

The data likelihood, $p(x|z)$ is known analytically, and $q(\beta|z)$ can be calculated according to Equation 13. The prior distribution, $p(\beta,z)$, is expressed in the following proposition:

**Proposition 5.1.** *Let $z$ be a set of topic assignments with integer values in $\{1,\ldots,K\}$. Let $\beta$ be a $(K+1)$–length vector representing global topic weights, and $\beta_{K+1}$ be the sum of weights associated with empty topics. The prior distribution, $p(\beta,z)$, marginalizing over $\pi$, can be expressed as*

$$p(\beta,z) = \left[\gamma\beta_{K+1}^{\gamma-1}\prod_{k=1}^{K}\beta_k^{-1}\right] \times \left[\prod_{j=1}^{D}\frac{\Gamma(\alpha)}{\Gamma(\alpha+n_{j\cdot\cdot})}\prod_{k=1}^{K}\frac{\Gamma(\alpha\beta_k+n_{j\cdot k})}{\Gamma(\alpha\beta_k)}\right]. \tag{15}$$

*Proof.* See supplemental material. □

The remaining term in Equation (14), $q(\hat{z}|v,\overline{v})$, is the probability of proposing a particular split. In the following sections, we describe three possible split constructions using the sub-clusters. Since the other steps remain the same, we only discuss the proposal distributions for $\hat{z}$ and $\hat{\beta}$.

### 5.1.1 Deterministic Split/Merge Proposals

The method of [1] constructs a split deterministically by copying the sub-cluster labels for a single cluster. We refer to this proposal as a *local* split, which only changes assignments within one topic, as opposed to a *global* split (discussed shortly), which changes all topic assignments. A local deterministic split will essentially be accepted if the joint likelihood increases. Unfortunately, as we show in the supplement, samples from the typical set of an HDP do not have high likelihood. Deterministic split and merge proposals are, consequently, very rarely accepted. We now suggest two alternative pairs of split and merge proposals, each with their own benefits and drawbacks.

### 5.1.2 Local Split/Merge Proposals

Here, we depart from the approach of [1] by *sampling* a local split of topic $a$ into topics $b$ and $c$. Temporary parameters, $\{\tilde{\pi}_b,\tilde{\pi}_c,\tilde{\theta}_b,\tilde{\theta}_c\}$, and topic assignments, $\hat{z}$, are sampled according to

$$\left.\begin{array}{l}(\tilde{\pi}_b,\tilde{\pi}_c) = \pi_a\cdot(\overline{\pi}_{a\ell},\overline{\pi}_{ar}),\\[4pt](\tilde{\theta}_b,\tilde{\theta}_c) = (\overline{\theta}_{a\ell},\overline{\theta}_{ar}),\end{array}\right\} \implies q(\hat{z}|v,\overline{v}) \propto \prod_{j,i\in\mathcal{I}_a}\sum_{k\in\{b,c\}}\tilde{\pi}_k f_x(x_{ji};\tilde{\theta}_k)\mathbb{1}[\hat{z}_{ji}=k]. \tag{16}$$

We note that a sample from $q(\hat{z}|v,\overline{v})$ is already drawn from the restricted Gibbs sampler described in Equation (9). Therefore, no additional computation is needed to propose the split. If the split is rejected, the $\hat{z}$ is simply used as the next sample of the auxiliary $\overline{z}$ for cluster $a$.

A $\hat{\beta}$ is then drawn by splitting $\beta_a$ into $\hat{\beta}_b$ and $\hat{\beta}_c$ according to a local version of Equation (13):

$$q(\hat{\beta}_b,\hat{\beta}_c|\hat{z},\beta_a) = \text{Dir}(\hat{\beta}_b/\beta_a,\hat{\beta}_c/\beta_a;\hat{\tilde{m}}_{\cdot b},\hat{\tilde{m}}_{\cdot c}). \tag{17}$$

The corresponding merge move combines topics $b$ and $c$ into topic $a$ by deterministically performing

$$q(\hat{z}_{ji}|v) = \mathbb{1}[\hat{z}_{ji}=a], \quad \forall j,i\in\mathcal{I}_b\cup\mathcal{I}_c, \qquad q(\hat{\beta}_a|v) = \delta(\hat{\beta}_a - (\beta_b+\beta_c)). \tag{18}$$

This results in the following Hastings ratio for a local split (derivation in supplement):

$$H = \frac{\gamma\Gamma(\hat{\tilde{m}}_{\cdot b})\Gamma(\hat{\tilde{m}}_{\cdot c})}{\Gamma(\hat{\tilde{m}}_{\cdot b}+\hat{\tilde{m}}_{\cdot c})}\frac{\beta_a^{\hat{\tilde{m}}_{\cdot b}+\hat{\tilde{m}}_{\cdot c}}}{\hat{\beta}_b^{\hat{\tilde{m}}_{\cdot b}}\hat{\beta}_c^{\hat{\tilde{m}}_{\cdot c}}}\frac{p(x|\hat{z})}{p(x|z)}\frac{1}{q(\hat{z}|v,\overline{v})}\frac{Q_{K+1}^M}{Q_K^S}\prod_j\frac{\Gamma(\alpha\beta_a)}{\Gamma(\alpha\beta_a+n_{j\cdot a})}\prod_{k\in\{b,c\}}\frac{\Gamma(\alpha\hat{\beta}_k+\hat{n}_{j\cdot k})}{\Gamma(\alpha\hat{\beta}_k)}, \tag{19}$$

where $Q_K^S$ and $Q_K^M$ are the probabilities of selecting a specific split or merge with $K$ topics. We record $q(\hat{z}|v,\overline{v})$ when sampling from Equation (9), and all other terms are computed via sufficient statistics. We set $Q_K^S = 1$ by proposing all splits at each iteration. $Q_K^M$ will be discussed shortly.

The Hastings ratio for a merge is essentially the reciprocal of Equation (19). However, the reverse split move, $q(z|\hat{v}, \hat{\bar{v}})$, relies on the inferred sub-topic parameters, $\hat{\bar{\pi}}$ and $\hat{\bar{\theta}}$, which are not readily available due to the Deferred MH algorithm. Instead, we approximate the Hastings ratio by substituting the two *original* topic parameters, $\theta_b$ and $\theta_c$, for the proposed sub-topics. The quality of this approximation rests on the similarity between the regular-topics and the sub-topics. Generating the reverse move that splits topic $a$ into $b$ and $c$ can then be approximated as

$$q(z|\hat{v}, \hat{\bar{v}}) \approx \prod_{j,i \in \mathcal{I}_b \cup \mathcal{I}_c} \frac{\pi_{z_{ji}} f_x(x_{ji};\theta_{z_{ji}})}{\pi_b f_x(x_i;\theta_b) + \pi_c f_x(x_i;\theta_c)} = \frac{\mathcal{L}_{bb}\mathcal{L}_{cc}}{\mathcal{L}_{bc}\mathcal{L}_{cb}}, \qquad (20)$$

$$\mathcal{L}_{kk} \triangleq \prod_{j,i \in \mathcal{I}_k} \pi_k f_x(x_{ji};\theta_k), \qquad \mathcal{L}_{kl} \triangleq \prod_{j,i \in \mathcal{I}_k} \left[\pi_k f_x(x_{ji};\theta_k) + \pi_l f_x(x_{ji};\theta_l)\right]. \qquad (21)$$

All of the terms in Equation (20) are already calculated in the restricted Gibbs steps. When aggregated correctly in the $K \times K$ matrix, $\mathcal{L}$, the Hastings ratio for any proposed merge is evaluated in constant time. However, if topics $b$ and $c$ are merged into $a$, further merging $a$ with another cluster cannot be efficiently computed without looping through the data. We therefore only propose $\lfloor K/2 \rfloor$ merges by generating a random permutation of the integers $[1, K]$, and proposing to merge disjoint neighbors. For example, if the random permutation for $K = 7$ is $\{\overline{3\,1}\ \overline{7\,4}\ \overline{2\,6}\ 5\}$, we propose to merge topics 3 and 1, topics 7 and 4, and topics 2 and 6. This results in $Q_K^M = \frac{2\lfloor K/2 \rfloor}{K(K-1)}$.

### 5.1.3 Global Split/Merge Proposals

In many applications where clusters have significant overlap (e.g., topic modeling), local splits may be too constrained since only points within a single topic change. We now develop a *global* split and merge move, which reassign the data in *all* topics. A global split first constructs temporary topic proportions, $\tilde{\pi}$, and parameters, $\tilde{\theta}$, followed by proposing topic assignments for *all* words with:

$$\left.\begin{array}{ll} (\tilde{\pi}_b, \tilde{\pi}_c) = \pi_a \cdot (\overline{\pi}_{a\ell}, \overline{\pi}_{ar}), & \tilde{\pi}_k = \pi_k, \quad \forall k \neq a, \\ (\tilde{\theta}_b, \tilde{\theta}_c) = (\overline{\theta}_{a\ell}, \overline{\theta}_{ar}), & \tilde{\theta}_k = \theta_k, \quad \forall k \neq a, \end{array}\right\} \Longrightarrow q(\hat{z}|v, \overline{v}) = \prod_{j,i} \frac{\tilde{\pi}_{\hat{z}_{ji}} f_x(x_{ji};\tilde{\theta}_{\hat{z}_{ji}})}{\sum_k \tilde{\pi}_k f_x(x_{ji};\tilde{\theta}_k)}. \quad (22)$$

Similarly, the corresponding merge move is constructed according to

$$\left.\begin{array}{ll} \tilde{\pi}_a = \pi_b + \pi_c, & \tilde{\pi}_k = \pi_k, \quad \forall k \neq b,c, \\ \tilde{\theta}_a \sim q(\tilde{\theta}_a|z,x), & \tilde{\theta}_k = \theta_k, \quad \forall k \neq b,c, \end{array}\right\} \Longrightarrow q(\hat{z}|v, \overline{v}) = \prod_{j,i} \frac{\tilde{\pi}_{\hat{z}_{ji}} f_x(x_{ji};\tilde{\theta}_{\hat{z}_{ji}})}{\sum_k \tilde{\pi}_k f_x(x_{ji};\tilde{\theta}_k)}. \quad (23)$$

The proposal for $\tilde{\theta}_a$ is written in a general form; if priors are conjugate, one should propose directly from the posterior. After Equations (22)–(23), $\hat{\beta}$ is sampled via Equation (13). All remaining steps follow Algorithm 1. The resulting Hastings ratio for a global split (see supplement) is expressed as

$$H = \frac{\gamma \Gamma(\gamma + \tilde{m}_{..})}{\Gamma(\gamma + \hat{m}_{..})} \frac{p(x|\hat{z})}{p(x|z)} \frac{q(z|\hat{v}, \hat{\bar{v}})q(\tilde{\theta}_a|z)}{q(\hat{z}|v, \overline{v})} \frac{Q_{K+1}^M}{Q_K^S} \prod_{k=1}^{K} \frac{\beta_k^{\bar{m}\cdot k}}{\Gamma(\tilde{m}_{\cdot k})} \prod_{j=1}^{D} \frac{\Gamma(\alpha\beta_k)}{\Gamma(\alpha\beta_k + n_{j\cdot k})} \prod_{k=1}^{K+1} \frac{\Gamma(\hat{m}_{\cdot k})}{\hat{\beta}_k^{\bar{m}\cdot k}} \prod_{j=1}^{D} \frac{\Gamma(\alpha\hat{\beta}_k + \hat{n}_{j\cdot k})}{\Gamma(\alpha\hat{\beta}_k)}. \quad (24)$$

Similar to local merges, the Hastings ratio for a global merge depends on the proposed sub-topics parameters. We approximate these with the main-topic parameters prior to the merge.

Unlike the local split/merge proposals, proposing $\hat{z}$ requires significant computation by looping through all data points. As such, we only propose a single global split and merge each iteration. Thus, $Q_K^S = 1/K$ and $Q_K^M = 2/(K(K-1))$. We emphasize that the developed *global* moves are very different from previous local split/merge moves in DPs and HDPs (e.g., [1, 7, 11, 13, 14]). We conjecture that this is the reason the split/merge moves in [7] only made negligible improvement.

## 6 Experiments

We now test the proposed HDP Sub-Clusters method on topic modeling. The algorithm is summarized in the following steps: **(1)** initialize $\beta$ and $z$ randomly; **(2)** sample $\pi$, $\theta$, $\overline{\pi}$, and $\overline{\theta}$ via Equations (2, 3, 7, 8); **(3)** sample $z$ and $\overline{z}$ via Equations (4, 9); **(4)** propose $\lfloor \frac{K}{2} \rfloor$ local merges followed by $K$ local splits; **(5)** propose a global merge followed by a global split; **(6)** sample $m$ and $\overline{m}$ via Equations (5, 10); **(7)** sample $\beta$ and $\overline{\beta}$ via Equations (1, 6); **(8)** repeat from Step 2 until convergence. We fix the hyper-parameters, but resampling techniques [2] can easily be incorporated. All results are averaged over 10 sample paths. Source code can be downloaded from http://people.csail.mit.edu/jchang7.

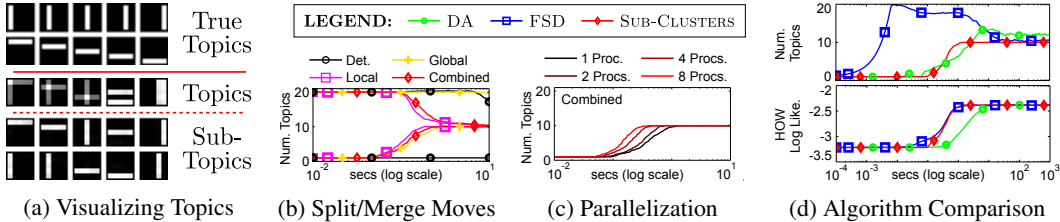

| (a) Visualizing Topics | (b) Split/Merge Moves | (c) Parallelization | (d) Algorithm Comparison |

Figure 3: Synthetic "bars" example. (a) Visualizing topic word distributions **without** splits/merges for $K = 5$. (b)–(c) Number of inferred topics for different split/merge proposals and parallelizations. (d) Comparing sampling algorithms with a single processor and initialized to a single topic.

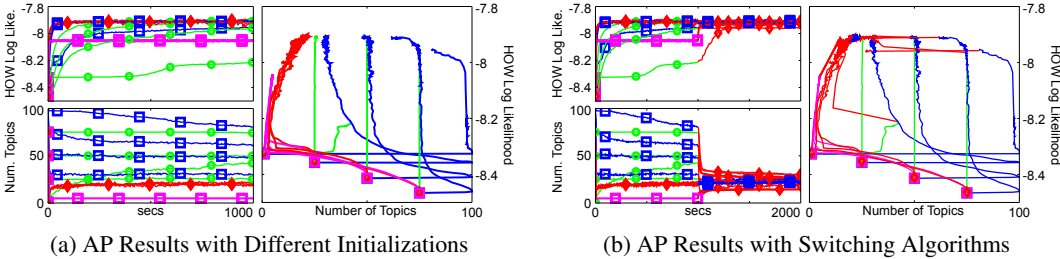

| (a) AP Results with Different Initializations | (b) AP Results with Switching Algorithms |

Figure 4: Results on AP. (a) 1, 25, 50, and 75 initial topics. (b) Switching algorithms at 1000 secs.

## 6.1 Synthetic Bars Dataset

We synthesized 200 documents from the "bars" example of [22] with a dictionary of 25 words that can be arranged in a 5x5 grid. Each of the 10 true topics forms a horizontal or vertical bar. To visualize the sub-topics, we initialize to 5 topics and **do not** propose splits or merges. The resulting regular- and sub-topics are shown in Figure 3a. Notice how the sub-topics capture likely splits.

Next, we consider different split/merge proposals in Figure 3b. The "Combined" algorithm uses local and global moves. The deterministic moves are often rejected resulting in slow convergence. While global moves are not needed in such a well-separated dataset, we have observed that the make a significant impact in real-world datasets. Furthermore, since every step of the sampling algorithm can be parallelized, we achieve a linear speedup in the number of processors, as shown in Figure 3c.

Figure 3d compares convergence without parallelization to the Direct Assignment (DA) sampler and the Finite Symmetric Dirichlet (FSD) of order 20. Since all algorithms should sample from the same model, the goal here is to analyze convergence *speed*. We plot two *summary statistics*: the likelihood of a single held-out word (HOW) from each document, and the number of inferred topics. While the HOW likelihood for FSD converges at 1 second, the number of topics converges at 100 seconds. This suggests that cross-validation techniques, which evaluate model fit, cannot solely determine MCMC convergence. We note that FSD tends to first create all $L$ topics and slowly remove them.

## 6.2 Real-World Corpora Datasets

Next, we consider the Associated Press (AP) dataset [23] with 436K words in 2K documents. We manually set the FSD order to 100. Results using 16 cores (except DA, which cannot be parallelized) with 1, 25, 50, and 75 initial topics are shown in Figure 4a. All samplers should converge to the same statistics regardless of the initialization. While HOW likelihood converges for 3/4 FSD initializations, the number of topics indicates that no DA or FSD sample paths have converged. Unlike the well-separated, synthetic dataset, the Sub-Clusters method that only uses local splits and merges does not converge to a good solution here. In contrast, all initializations of the Sub-Clusters method have converged to a high HOW likelihood with only approximately 20 topics. The *path* taken by each sampler in the joint HOW likelihood / number of topics space is shown in the right panel of Figure 4a. This visualization helps to illustrate the different approaches taken by each algorithm.

Figure 5a shows *confusion matrices*, $C$, of the inferred topics. Each element of $C$ is defined as: $C_{r,c} = \sum_x f_x(x; \theta_r) \log f_x(x; \theta_c)$, and captures the likelihood of a random word from topic $r$

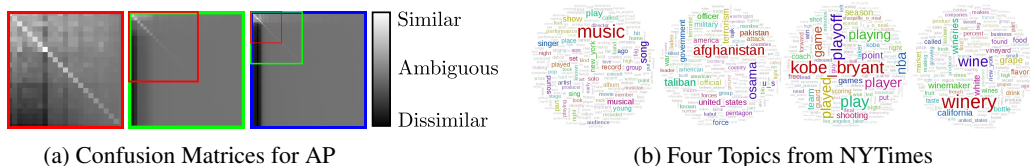

| (a) Confusion Matrices for AP | (b) Four Topics from NYTimes |
|---|---|

Figure 5: (a) Confusion matrices on AP for SUB-CLUSTERS, DA, and FSD (left to right). Outlines are overlaid to compare size. (b) Four inferred topics from the NYTimes articles.

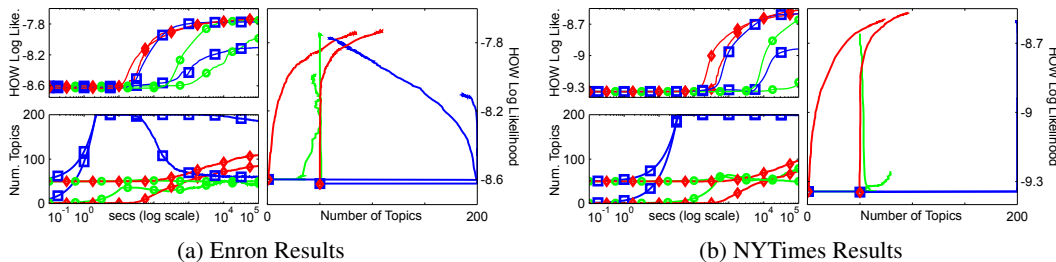

| (a) Enron Results | (b) NYTimes Results |
|---|---|

Figure 6: Results on (a) Enron emails and (b) NYTimes articles for 1 and 50 initial topics.

evaluated under topic $c$. DA and FSD both converge to many topics that are easily confused, whereas the Sub-Clusters method converges to a smaller set of more distinguishable topics.

Rigorous proofs about convergence are quite difficult. Furthermore, even though the approximations made in calculating the Hastings ratios for local and global splits (e.g., Equation (20)) are backed by intuition, they complicate the analysis. Instead, we run each sample path for 2,000 seconds. After 1,000 seconds, we switch the Sub-Clusters sample paths to FSD and all other sample paths to Sub-Clusters. Markov chains that have converged should not change when switching the sampler. Figure 4b shows that switching from DA, FSD, or the local version of Sub-Clusters immediately changes the number of topics, but switching Sub-Clusters to FSD has no effect. We believe that the number of topics is slightly higher in the former because the Sub-Cluster method struggles to create small topics. By construction, the splits make large moves, in contrast to DA and FSD, which often create single word topics. This suggests that alternating between FSD and Sub-Clusters may work well.

Finally, we consider two large datasets from [24]: Enron Emails with 6M words in 40K documents and NYTimes Articles with 100M words in 300K documents. We note that the NYTimes dataset is 3 orders of magnitude larger than those considered in the HDP split/merge work of [7]. Again, we manually set the FSD order to 200. Results are shown in Figure 6 initialized to 1 and 50 topics. In such large datasets, it is difficult to predict convergence times; after 28 hours, it seems as though no algorithms have converged. However, the Sub-Clusters method seems to be approaching a solution, whereas FSD has yet to prune topics and DA has yet to to achieve a good cross-validation score. Four inferred topics using the Sub-Clusters method on the NYTimes dataset are visualized in Figure 5b. These words seem to describe plausible topics (e.g., music, terrorism, basketball, and wine).

## 7 Conclusion

We have developed a new parallel sampling algorithm for the HDP that proposes split and merge moves. Unlike previous attempts, the proposed *global* splits and merges exhibit significantly improved convergence in a variety of datasets. We have also shown that cross-validation metrics in isolation can lead to the erroneous conclusion that an MCMC sampling algorithm has converged. By considering the number of topics and held-out likelihood jointly, we show that previous sampling algorithms converge very slowly.

**Acknowledgments**

This research was partially supported by the Office of Naval Research Multidisciplinary Research Initiative program, award N000141110688 and by VITALITE, which receives support from Army Research Office Multidisciplinary Research Initiative program, award W911NF-11-1-0391.

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
