[Supplementary Material]

# Supplemental Material for
# *Parallel Sampling of HDPs using Sub-Cluster Splits*

**Jason Chang**
CSAIL, MIT
jchang7@csail.mit.edu

**John W. Fisher III**
CSAIL, MIT
fisher@csail.mit.edu

In the following supplemental material we provide some additional details and derivations for the paper. We begin by showing how to calculate the joint distribution of $p(\beta, z)$, marginalizing out $\pi$, in Section 1. Then, in Section 2 we consider looking at joint log-likelihoods of HDP topic models and show that the typical set of the distribution is very far from the mode. In Sections 3-4, we give a more detailed calculation of the Hastings ratios. Finally, in Section 5, we visualize the inferred topics from the New York Times Articles.

## 1 Calculating the $p(\beta, z)$ Distribution

We begin by proving Proposition 5.1 in the paper, reproduced here.

**Proposition 1.1.** *Let $z$ be a set of topic assignments with integer values in $\{1, \dots, K\}$. Let $\beta$ be a $(K+1)$–length vector representing global topic weights, and $\beta_{K+1}$ be the sum of weights associated with empty topics. The prior distribution, $p(\beta, z)$, marginalizing over $\pi$, can be expressed as*

$$p(\beta, z) = \left[ \gamma \beta_{K+1}^{\gamma-1} \prod_{k=1}^{K} \beta_k^{-1} \right] \times \left[ \prod_{j=1}^{D} \frac{\Gamma(\alpha)}{\Gamma(\alpha+n_{j\cdot\cdot})} \prod_{k=1}^{K} \frac{\Gamma(\alpha\beta_k+n_{j\cdot k})}{\Gamma(\alpha\beta_k)} \right]. \tag{1}$$

To calculate the prior distribution, $p(\beta, z)$, we will rely heavily on the Chinese Restaurant Franchise (CRF) representation of the HDP given in [1]. Additionally, we change the notation slightly. In [1], $t$ was used as a particular table index, and $t_{ji}$ was used as the particular index assigned to customer $i$ in restaurant $j$. This can be confusing, since we often refer to a variable without subscripts as the set of all variables. Instead, we denote $\tau_{ji}$ as the table assignment for customer $i$ in restaurant $j$, and $\kappa_{jt}$ as the dish assignment for table $t$ in restaurant $j$. The derivation is outlined in the following steps.

1. Find $p(\beta, \kappa, \tau, z)$
2. Find $p(\kappa, \tau | \beta, z)$
3. Combine to find $p(\beta, z)$

### 1.1 Deriving the Joint: $p(\beta, \kappa, \tau, z)$

Based on the generative process of the CRF, $p(\kappa, \tau)$ can be expressed as

$$p(\tau) = \prod_{j=1}^{D} \text{CRP}(\alpha, n_{j\cdot\cdot}) = \prod_{j=1}^{D} \frac{\Gamma(\alpha)\alpha^{m_{j\cdot}}}{\Gamma(\alpha + n_{j\cdot\cdot})} \prod_{t=1}^{m_{j\cdot}} \Gamma(n_{jt\cdot}), \tag{2}$$

$$p(\kappa|\tau) = \text{CRP}(\gamma, m_{\cdot\cdot}) = \frac{\Gamma(\gamma)\gamma^{K}}{\Gamma(\gamma + m_{\cdot\cdot})} \prod_{k=1}^{K} \Gamma(m_{\cdot k}), \tag{3}$$

where CRP$(\cdot)$ represents a sample from a Chinese Restaurant Process. These expressions can be combined to form the joint:

$$p(\kappa, \tau) = \left[ \frac{\Gamma(\gamma)\gamma^K}{\Gamma(\gamma + m_{..})} \prod_{k=1}^{K} \Gamma(m_{\cdot k}) \right] \cdot \left[ \prod_{j=1}^{D} \frac{\Gamma(\alpha)\alpha^{m_{j\cdot}}}{\Gamma(\alpha + n_{j..})} \prod_{t=1}^{m_{j\cdot}} \Gamma(n_{jt\cdot}) \right]. \tag{4}$$

We note that $z$, which directly assigns a customer to a dish, is a deterministic function conditioned on $\kappa$ and $\tau$. More precisely, it can be expressed as

$$p(z|\kappa, \tau) = \prod_{j=1}^{D} \prod_{i=1}^{N_j} \mathbb{1}[z_{ji} = \kappa_{j\tau_{ji}}]. \tag{5}$$

Additionally, it is well known that $p(\beta|m)$ is the following Dirichlet distribution

$$p(\beta|m) = \text{Dir}(\beta_1, \ldots, \beta_K, \beta_{K+1}; m_{\cdot 1}, \cdots, m_{\cdot K}, \gamma). \tag{6}$$

This can be seen by drawing on the fact that any partitioning of the space in a Dirichlet process results in a Dirichlet distribution. Since $m$ is a summary statistic of $k$, we have

$$p(\beta|\kappa, \tau, z) = p(\beta|m) = \frac{\Gamma(\gamma + m_{..})\beta_{K+1}^{\gamma - 1}}{\Gamma(\gamma) \prod_{k=1}^{K} \Gamma(m_{\cdot k})} \prod_{k=1}^{K} \beta_k^{m_{\cdot k} - 1} \tag{7}$$

Finally, assuming $z$ is consistent with $\kappa$ and $\tau$ (i.e., Equation (5) evaluates to 1 instead of 0), the entire joint prior of interest can be expressed as

$$p(\beta, \kappa, \tau, z)$$

$$= \left[ \frac{\Gamma(\gamma)\gamma^K}{\Gamma(\gamma + m_{..})} \prod_{k=1}^{K} \Gamma(m_{\cdot k}) \right] \left[ \prod_{j=1}^{D} \frac{\Gamma(\alpha)\alpha^{m_{j\cdot}}}{\Gamma(\alpha + n_{j..})} \prod_{t=1}^{m_{j\cdot}} \Gamma(n_{jt\cdot}) \right] \left[ \frac{\Gamma(\gamma + m_{..})\beta_{K+1}^{\gamma - 1}}{\Gamma(\gamma) \prod_{k=1}^{K} \Gamma(m_{\cdot k})} \prod_{k=1}^{K} \beta_k^{m_{\cdot k} - 1} \right]$$

$$= \gamma^K \beta_{K+1}^{\gamma - 1} \alpha^{m_{..}} \left[ \prod_{k=1}^{K} \beta_k^{m_{\cdot k} - 1} \right] \left[ \prod_{j=1}^{D} \frac{\Gamma(\alpha)}{\Gamma(\alpha + n_{j..})} \prod_{t=1}^{m_{j\cdot}} \Gamma(n_{jt\cdot}) \right] \tag{8}$$

## 1.2 Deriving the Conditional $p(\kappa, \tau|\beta, z)$

We now show how to express $p(\kappa, \tau|\beta, z)$. We note that conditioning on $z$ is equivalent to assigning each customer a particular dish. Thus, we need to calculate the probability of any particular configuration of tables such that each customer gets the correct dish.

Suppose there are three customers with assignments $z_{11} = 1$, $z_{12} = 2$, and $z_{13} = 2$. While $x_{11}$ must sit at a different table than $x_{12}$ and $x_{13}$ (i.e., $\tau_{12} \neq \tau_{11} \neq \tau_{23}$), nothing can be said about the relationship between $\tau_{12}$ and $\tau_{13}$. This results from the fact that two customers can be served the same dish at different tables.

An equivalent metaphor for the process conditioned on $z$ is that a customer comes into a restaurant having been assigned dish $k$. The customer then chooses to sit at an occupied table serving dish $k$ with probability proportional to the number of customers there, or starts a new table that serves dish $k$ with probability $\alpha\beta_k$. This process is equivalent to $D \times K$ independent CRPs, each with $n_{j\cdot k}$ customers and $\alpha\beta_k$ as the concentration parameter. Thus, we can write this easily as

$$p(\kappa, \tau|\beta, z) = \prod_{j=1}^{D} \prod_{k=1}^{K} \text{CRP}(\alpha\beta_k, n_{j\cdot k}) \tag{9}$$

$$= \prod_{j=1}^{D} \prod_{k=1}^{K} \frac{(\alpha\beta_k)^{m_{jk}} \Gamma(\alpha\beta_k)}{\Gamma(\alpha\beta_k + n_{j\cdot k})} \prod_{t=1}^{m_{jk}} \Gamma(n_{jtk}) \tag{10}$$

$$= \alpha^{m_{..}} \left[ \prod_{j=1}^{D} \prod_{k=1}^{K} \frac{\beta_k^{m_{jk}} \Gamma(\alpha\beta_k)}{\Gamma(\alpha\beta_k + n_{j\cdot k})} \right] \left[ \prod_{j=1}^{D} \prod_{t=1}^{m_{jk}} \Gamma(n_{jt\cdot}) \right], \tag{11}$$

where we have used the fact that every table only serves one dish to equate the following

$$\prod_{k=1}^{K} \prod_{t=1}^{m_{jk}} \Gamma(n_{jtk}) = \prod_{t=1}^{m_{j\cdot}} \Gamma(n_{jt\cdot}). \tag{12}$$

### 1.3 Finding the Prior $p(\beta, z)$

We now note the following relationship:

$$p(\beta, \kappa, \tau, z) = p(\beta, z)p(\kappa, \tau | \beta, z). \tag{13}$$

Finding the expression for $p(\beta, z)$ is as simple as substituting the previously found expressions. Assuming consistency between $z$ with $\kappa$ and $\tau$, we can ignore the $p(z | \kappa, \tau)$ term in Equation (5), resulting in

$$p(\beta, z) = \frac{p(\beta, \kappa, \tau, z)}{p(\kappa, \tau | \beta, z)}, \tag{14}$$

$$= \frac{\gamma^K \beta_{K+1}^{\gamma-1} \alpha^{m_{\cdot\cdot}} \left[ \prod_{k=1}^{K} \beta_k^{m_{\cdot k}-1} \right] \left[ \prod_{j=1}^{D} \frac{\Gamma(\alpha)}{\Gamma(\alpha+n_{j\cdot\cdot})} \prod_{t=1}^{m_{j\cdot}} \Gamma(n_{jt\cdot}) \right]}{\alpha^{m_{\cdot\cdot}} \left[ \prod_{j=1}^{D} \prod_{k=1}^{K} \frac{\Gamma(\alpha\beta_k)}{\Gamma(\alpha\beta_k+n_{j\cdot k})} \beta_k^{m_{jk}} \right] \left[ \prod_{j=1}^{D} \prod_{t=1}^{m_{j\cdot}} \Gamma(n_{jt\cdot}) \right]}, \tag{15}$$

$$= \frac{\gamma^K \beta_{K+1}^{\gamma-1} \left[ \prod_{k=1}^{K} \beta_k^{-1} \right] \left[ \prod_{j=1}^{D} \frac{\Gamma(\alpha)}{\Gamma(\alpha+n_{j\cdot\cdot})} \right]}{\prod_{j=1}^{D} \prod_{k=1}^{K} \frac{\Gamma(\alpha\beta_k)}{\Gamma(\alpha\beta_k+n_{j\cdot k})}}, \tag{16}$$

$$= \gamma^K \beta_{K+1}^{\gamma-1} \prod_{k=1}^{K} \beta_k^{-1} \left[ \prod_{j=1}^{D} \frac{\Gamma(\alpha)}{\Gamma(\alpha+n_{j\cdot\cdot})} \prod_{k=1}^{K} \frac{\Gamma(\alpha\beta_k+n_{j\cdot k})}{\Gamma(\alpha\beta_k)} \right]. \tag{17}$$

This concludes the derivation of finding the expression for $p(\beta, z)$.

### 1.4 Notes on $p(\beta, z)$

We highlight a few notes on the derived expression for $p(\beta, z)$. At first glance, parts of Equation (17) may seem a bit odd. For example, the $\beta_k^{-1}$ term seems like an improper prior, and the term inside the square brackets just seems like the product of Dirichlet-Multinomial distributions. We remind the reader that meaning of $\beta$ in $p(\beta, z)$ slightly differs from the infinite-length global topic proportions. In particular, $\beta$ is defined over the partitions imposed by $z$. Moreover, because $z$ takes on exactly $K$ non-empty partitions, $\beta_k > 0 \ \forall k \in \{1, \ldots, K\}$, and $\beta_k^{-1}$ will never result in division by zero.

The term in the square brackets is very similar to a Dirichlet-Multinomial. A product of $D$ independent Dirichlet-Multinomial distributions can be expressed as

$$\prod_{j=1}^{D} \text{Dir-Mult}(z_j; \alpha\beta_1, \ldots, \alpha\beta_{K+1}) = \prod_{j=1}^{D} \frac{\Gamma(\alpha)}{\Gamma(\alpha+n_{j\cdot\cdot})} \prod_{k=1}^{K+1} \frac{\Gamma(\alpha\beta_k+n_{j\cdot k})}{\Gamma(\alpha\beta_k)},$$

where we have used the fact that $\sum_{k=1}^{K} \alpha\beta_k = \alpha$. While this expression looks similar to the one in Equation (17), the inner product is over $K + 1$ terms instead of $K$. Another way of viewing the difference is that Equation (17) implicitly assumes that $n_{j\cdot(K+1)}$ is zero, since $z$ can only take on $K$ unique partitions. For this reason, the term inside the square brackets of Equation (17) is not the product of Dirichlet-Multinomials.

Equation (17) cannot be analytically integrated to validate that it has the correct normalization. We know, however, that Equation (11) describing $p(\kappa, \tau | \beta, z)$ trivially integrates to one by the construction of the independent CRPs. Because $p(\kappa, \tau | \beta, z)$ is a valid distribution and $p(\beta, \kappa, \tau, z)/p(\kappa, \tau | \beta, z)$ has no dependence on $\kappa$ or $\tau$, the derived expression for $p(\beta, z)$ must be a valid distribution, conditioned on $p(\beta, \kappa, \tau, z)$ being the correct joint distribution.

Figure 1: Joint posterior model log likelihood plotted against number of topics for the Associated Press dataset.

## 2 Joint Model Likelihoods

When the desired topic distributions have a lot of overlap, the resulting likelihood of a sample from the typical set under the posterior is typically much smaller than the mode of the distribution. To illustrate this observation, we consider the Associated Press dataset of [2]. We initialize an HDP sample with $K$ initial topics, and do not allow the addition of new topics. The resulting joint posterior log likelihood for the entire model is shown in Figure 1 Clearly, the joint likelihood is highest for one giant topic. However, we do not expect a sample from the posterior to only contain one topic. In other words, this means the configuration with a single topic is not in the typical set of the posterior, even though it has high likelihood.

For this reason, a deterministic split proposal that has $\frac{q(z|\hat{v},\tilde{\hat{v}})}{q(\hat{z}|v,\overline{v})}$ close to unity will almost always reject the sample, since the model likelihood decreases. This is slightly abnormal since, in general, proposals for a split will have the ratio, $\frac{q(z|\hat{v},\tilde{\hat{v}})}{q(\hat{z}|v,\overline{v})}$, evaluate to much less than unity due to the fact that there is only one way to merge two clusters, but many ways to split a cluster into two. As such, the deterministic split proposals described in [3] do not work well in HDPs, and led us to develop the other local and global split/merge proposals.

## 3 Hastings Ratios for Local Proposals

For the proposed local split of topic $a$, the only variables that are changed are $\beta_a$ and $z_{ji}$ for all points with label $a$. Thus, the ratio of posteriors for the local split can be expressed as

$$\frac{p(\hat{\beta}, \hat{z}, x)}{p(\beta, z, x)} = \frac{\gamma\beta_a}{\hat{\beta}_b\hat{\beta}_c} \left[ \prod_{j=1}^{D} \frac{\Gamma(\alpha\beta_a)}{\Gamma(\alpha\beta_a + n_{j\cdot a})} \prod_{k\in\{b,c\}} \frac{\Gamma(\alpha\hat{\beta}_k + \hat{n}_{j\cdot k})}{\Gamma(\alpha\hat{\beta}_k)} \right] \frac{p(x|\hat{z})}{p(x|z)} \tag{18}$$

Consider the proposal over the main variables $\hat{\beta}$ and $\hat{z}$. A split move is proposed as follows. Conditioned on $\beta$ and $z$, we propose a new $\hat{\beta}$ and $\hat{z}$ with the following:

$$\hat{z} \sim q(\hat{z}|v,\overline{v}) \tag{19}$$

$$(\tilde{\beta}_b, \tilde{\beta}_c) \sim \text{Dir}(\hat{\tilde{m}}_{\cdot b}(\hat{z}), \hat{\tilde{m}}_{\cdot c}(\hat{z})) \tag{20}$$

$$(\hat{\beta}_b, \hat{\beta}_c) = \beta_a \cdot (\tilde{\beta}_b, \tilde{\beta}_c), \tag{21}$$

where $q(\hat{z}|v,\overline{v})$ is described in the main paper. We are a little more careful with notation here, and denote $\hat{\tilde{m}}$ as the proposed temporary variables. Here, we considering calculating the proposal ratio for the $\beta$'s. We use the reversible jump algorithm [4] to calculate the ratio. The function $f$ maps us from

$$[\beta_a, \overline{\beta}_{a\ell}] \rightarrow [\hat{\beta}_b, \hat{\beta}_c] \tag{22}$$

The Jacobian matrix for the mapping is then

$$J_\beta = \begin{bmatrix} \frac{\partial \hat{\beta}_b}{\partial \beta_a} & \frac{\partial \hat{\beta}_b}{\partial \overline{\beta}_{a\ell}} \\ \frac{\partial \hat{\beta}_c}{\partial \beta_a} & \frac{\partial \hat{\beta}_c}{\partial \overline{\beta}_{a\ell}} \end{bmatrix} = \begin{bmatrix} \overline{\beta}_{a\ell} & \beta_a \\ (1 - \overline{\beta}_{a\ell}) & -\beta_a \end{bmatrix}, \tag{23}$$

which has an absolute value determinant of

$$|\det(J_\beta)| = \left| \beta_a \cdot \overline{\beta}_{a\ell} - \beta_a(1 - \overline{\beta}_{a\ell}) \right| = \beta_a. \tag{24}$$

The ratio of proposals is then just $\beta_a$ for a split proposal can then be expressed as:

$$\frac{q(\beta | z, \hat{v}, \hat{\overline{v}})}{q(\hat{\beta} | \hat{z}, v, \overline{v})} = \frac{\beta_a}{\mathrm{Dir}(\hat{\beta}_b/\beta_a, \hat{\beta}_c/\beta_a; \hat{m}_{\cdot b}, \hat{m}_{\cdot c})} \tag{25}$$

$$= \beta_a \frac{\Gamma(\hat{\tilde{m}}_{\cdot b})\Gamma(\hat{\tilde{m}}_{\cdot c})}{\Gamma(\hat{\tilde{m}}_{\cdot b} + \hat{\tilde{m}}_{\cdot c})} \left( \frac{\hat{\beta}_b}{\beta_a} \right)^{1 - \hat{\tilde{m}}_{\cdot b}} \left( \frac{\hat{\beta}_c}{\beta_a} \right)^{1 - \hat{\tilde{m}}_{\cdot c}} \tag{26}$$

$$= \frac{\Gamma(\hat{\tilde{m}}_{\cdot b})\Gamma(\hat{\tilde{m}}_{\cdot c})}{\Gamma(\hat{\tilde{m}}_{\cdot b} + \hat{\tilde{m}}_{\cdot c})} \frac{\hat{\beta}_b^{1 - \hat{\tilde{m}}_{\cdot b}} \hat{\beta}_c^{1 - \hat{\tilde{m}}_{\cdot c}}}{\beta_a^{1 - \hat{\tilde{m}}_{\cdot b} - \hat{\tilde{m}}_{\cdot c}}} \tag{27}$$

Combining these expressions results in the following Hastings ratio for a local split

$$H_S = \frac{p(\hat{\beta}, \hat{z}, x)}{p(\beta, z, x)} \frac{Q_{K+1}^M}{Q_K^S} \frac{q(\beta | z, \hat{v}, \hat{\overline{v}})}{q(\hat{\beta} | \hat{z}, v, \overline{v})} \frac{1}{q(\hat{z} | v, \overline{v})} \tag{28}$$

$$= \frac{\gamma \beta_a}{\hat{\beta}_b \hat{\beta}_c} \left[ \prod_{j=1}^{D} \frac{\Gamma(\alpha \beta_a)}{\Gamma(\alpha \beta_a + n_{j \cdot a})} \prod_{k \in \{b,c\}} \frac{\Gamma(\alpha \hat{\beta}_k + \hat{n}_{j \cdot k})}{\Gamma(\alpha \hat{\beta}_k)} \right] \frac{p(x|\hat{z})}{p(x|z)}$$

$$\times \frac{Q_{K+1}^M}{Q_K^S} \frac{\Gamma(\hat{\tilde{m}}_{\cdot b})\Gamma(\hat{\tilde{m}}_{\cdot c})}{\Gamma(\hat{\tilde{m}}_{\cdot b} + \hat{\tilde{m}}_{\cdot c})} \frac{\hat{\beta}_b^{1 - \hat{\tilde{m}}_{\cdot b}} \hat{\beta}_c^{1 - \hat{\tilde{m}}_{\cdot c}}}{\beta_a^{1 - \hat{\tilde{m}}_{\cdot b} - \hat{\tilde{m}}_{\cdot c}}} \frac{1}{q(\hat{z} | v, \overline{v})} \tag{29}$$

$$= \frac{\gamma \Gamma(\hat{\tilde{m}}_{\cdot b})\Gamma(\hat{\tilde{m}}_{\cdot c})}{\Gamma(\hat{\tilde{m}}_{\cdot b} + \hat{\tilde{m}}_{\cdot c})} \frac{\beta_a^{\hat{\tilde{m}}_{\cdot b} - \hat{\tilde{m}}_{\cdot c}}}{\hat{\beta}_b^{\hat{\tilde{m}}_{\cdot b}} \hat{\beta}_c^{\hat{\tilde{m}}_{\cdot c}}} \frac{p(x|\hat{z})}{p(x|z)} \frac{1}{q(\hat{z} | v, \overline{v})} \frac{Q_{K+1}^M}{Q_K^S} \prod_{j=1}^{D} \frac{\Gamma(\alpha \beta_a)}{\Gamma(\alpha \beta_a + n_{j \cdot a})} \prod_{k \in \{b,c\}} \frac{\Gamma(\alpha \hat{\beta}_k + \hat{n}_{j \cdot k})}{\Gamma(\alpha \hat{\beta}_k)}, \tag{30}$$

which matches Equation 19 in the main paper. Similarly, it can be shown that the Hastings ratio for a local merge is

$$H_M = \frac{\Gamma(\tilde{m}_{\cdot b} + \tilde{m}_{\cdot c})}{\gamma \Gamma(\tilde{m}_{\cdot b})\Gamma(\tilde{m}_{\cdot c})} \frac{\beta_b^{\tilde{m}_{\cdot b}} \beta_c^{\tilde{m}_{\cdot c}}}{\hat{\beta}_a^{\tilde{m}_{\cdot b} - \tilde{m}_{\cdot c}}} \frac{p(x|z)}{p(x|\hat{z})} q(\hat{z} | v, \overline{v}) \frac{Q_{K-1}^S}{Q_K^M} \prod_{j=1}^{D} \frac{\Gamma(\alpha \hat{\beta}_a + n_{j \cdot a})}{\Gamma(\alpha \hat{\beta}_a)} \prod_{k \in \{b,c\}} \frac{\Gamma(\alpha \beta_k)}{\Gamma(\alpha \beta_k + n_{j \cdot k})}, \tag{31}$$

where $\tilde{m}$ is a function of the original $z$.

## 4 Hastings Ratios for Global Proposals

For the proposed global split of topic $a$, all $\beta$'s and $z$'s change. Instead of denoting the empty $\beta$ with $\beta_{K+1}$, we use the notation $\beta_E$ here. The ratio of posteriors for the global split can be expressed as

$$\frac{p(\hat{\beta}, \hat{z}, x)}{p(\beta, z, x)} = \left[ \frac{\hat{\beta}_E}{\beta_E} \right]^{\gamma - 1} \frac{p(x|\hat{z})}{p(x|z)} \gamma \prod_{k=1}^{K} \beta_k \prod_{k=1}^{K+1} \hat{\beta}_k^{-1} \prod_{j=1}^{D} \prod_{k=1}^{K} \frac{\Gamma(\alpha \beta_k)}{\Gamma(\alpha \beta_k + n_{j \cdot k})} \prod_{k=1}^{K+1} \frac{\Gamma(\alpha \hat{\beta}_k + \hat{n}_{j \cdot k})}{\Gamma(\alpha \hat{\beta}_k)} \tag{32}$$

The proposal ratio for the $z$'s is

$$\frac{q(z | \hat{v}, \hat{\overline{v}})}{q(\hat{z} | v, \overline{v})} q(\tilde{\theta}_a | z) \tag{33}$$

We follow in the same steps as the previous local proposals by calculating the proposal ratio for the $\beta$'s.

$$\frac{q(\beta|z)}{q(\hat{\beta}|\hat{z})} = \frac{\text{Dir}(\beta_1,\ldots,\beta_K,\beta_E;\tilde{m}_{\cdot 1},\ldots,\tilde{m}_{\cdot K},\gamma)}{\text{Dir}(\hat{\beta}_1,\ldots,\hat{\beta}_{K+1},\hat{\beta}_E;\hat{\tilde{m}}_{\cdot 1},\ldots,\hat{\tilde{m}}_{\cdot(K+1)},\gamma)} \tag{34}$$

$$= \frac{\Gamma(\gamma+\tilde{m}_{\cdot\cdot})}{\Gamma(\gamma+\hat{\tilde{m}}_{\cdot\cdot})}\left(\frac{\beta_E}{\hat{\beta}_E}\right)^{\gamma-1}\prod_{k=1}^{K}\frac{\beta_k^{\tilde{m}_{\cdot k}-1}}{\Gamma(\tilde{m}_{\cdot k})}\prod_{k=1}^{K+1}\frac{\Gamma(\hat{\tilde{m}}_{\cdot k})}{\hat{\beta}_k^{\hat{\tilde{m}}_{\cdot k}-1}} \tag{35}$$

Combining these expressions, we arrive at the following Hastings ratio for a global split

$$
\begin{aligned}
H_S &= \frac{p(\hat{\beta},\hat{z},x)}{p(\beta,z,x)}\frac{Q_{K+1}^M}{Q_K^S}\frac{q(\beta|z,\hat{v},\tilde{\overline{v}})}{q(\hat{\beta}|\hat{z},v,\overline{v})}\frac{q(z|\hat{v},\tilde{\overline{v}})}{q(\hat{z}|v,\overline{v})}q(\tilde{\theta}_a|z)\\
&= \left[\frac{\hat{\beta}_E}{\beta_E}\right]^{\gamma-1}\frac{p(x|\hat{z})}{p(x|z)}\gamma\prod_{k=1}^{K}\beta_k\prod_{k=1}^{K+1}\hat{\beta}_k^{-1}\prod_{j=1}^{D}\prod_{k=1}^{K}\frac{\Gamma(\alpha\beta_k)}{\Gamma(\alpha\beta_k+n_{j\cdot k})}\prod_{k=1}^{K+1}\frac{\Gamma(\alpha\hat{\beta}_k+\hat{n}_{j\cdot k})}{\Gamma(\alpha\hat{\beta}_k)}\\
&\quad\times\frac{Q_{K+1}^M}{Q_K^S}\frac{q(z|\hat{v},\tilde{\overline{v}})}{q(\hat{z}|v,\overline{v})}q(\tilde{\theta}_a|z)\frac{\Gamma(\gamma+\tilde{m}_{\cdot\cdot})}{\Gamma(\gamma+\hat{\tilde{m}}_{\cdot\cdot})}\left(\frac{\beta_E}{\hat{\beta}_E}\right)^{\gamma-1}\prod_{k=1}^{K}\frac{\beta_k^{\tilde{m}_{\cdot k}-1}}{\Gamma(\tilde{m}_{\cdot k})}\prod_{k=1}^{K+1}\frac{\Gamma(\hat{\tilde{m}}_{\cdot k})}{\hat{\beta}_k^{\hat{\tilde{m}}_{\cdot k}-1}}\\
&= \frac{\gamma\Gamma(\gamma+\tilde{m}_{\cdot\cdot})}{\Gamma(\gamma+\hat{\tilde{m}}_{\cdot\cdot})}\frac{p(x|\hat{z})}{p(x|z)}\frac{q(z|\hat{v},\tilde{\overline{v}})}{q(\hat{z}|v,\overline{v})}\frac{q(\tilde{\theta}_a|x,z)}{1}\frac{Q_{K+1}^M}{Q_K^S}\\
&\quad\times\prod_{k=1}^{K}\frac{\beta_k^{\tilde{m}_{\cdot k}}}{\Gamma(\tilde{m}_{\cdot k})}\prod_{j=1}^{D}\frac{\Gamma(\alpha\beta_k)}{\Gamma(\alpha\beta_k+n_{j\cdot k})}\times\prod_{k=1}^{K+1}\frac{\Gamma(\hat{\tilde{m}}_{\cdot k})}{\hat{\beta}_k^{\hat{\tilde{m}}_{\cdot k}}}\prod_{j=1}^{D}\frac{\Gamma(\alpha\hat{\beta}_k+\hat{n}_{j\cdot k})}{\Gamma(\alpha\hat{\beta}_k)},
\end{aligned} \tag{36}
$$

which matches Equation 24 in the main paper. Similarly, it can be shown that the Hastings ratio for a global merge is

$$
\begin{aligned}
H_M &= \frac{\Gamma(\gamma+\tilde{m}_{\cdot\cdot})}{\gamma\Gamma(\gamma+\hat{\tilde{m}}_{\cdot\cdot})}\frac{p(x|\hat{z})}{p(x|z)}\frac{q(z|\hat{v},\tilde{\overline{v}})}{q(\hat{z}|v,\overline{v})}\frac{1}{q(\tilde{\theta}_c|x,\hat{z})}\frac{Q_{K-1}^S}{Q_K^M}\\
&\quad\times\prod_{k=1}^{K}\frac{\beta_k^{\tilde{m}_{\cdot k}(z)}}{\Gamma(\tilde{m}_{\cdot k}(z))}\prod_{j=1}^{D}\frac{\Gamma(\alpha\beta_k)}{\Gamma(\alpha\beta_k+n_{j\cdot k})}\times\prod_{k=1}^{K-1}\frac{\Gamma(\hat{\tilde{m}}_{\cdot k})}{\hat{\beta}_k^{\hat{\tilde{m}}_{\cdot k}}}\prod_{j=1}^{D}\frac{\Gamma(\alpha\hat{\beta}_k+\hat{n}_{j\cdot k})}{\Gamma(\alpha\hat{\beta}_k)},
\end{aligned} \tag{37}
$$

## 5 Topic Visualization

The word clouds at the end of this document summarize the inferred topics for a single sample path in the dataset of New York Times Articles. The topics are ordered in decreasing values of $\beta$, with highest likely topics appearing first.