[Reviews · NeurIPS 2014]

Submitted by Assigned_Reviewer_13

The authors present a sampling technique for Hierarchical Dirichlet process models which accelerates split-merge moves by augmenting the parameter space to include sub-clusters. This implies that as the standard parameters are learned, at the same time parameters for cluster splits are learned which can be used to propose sensible split moves, both local and global.

The concept of sub-clusters appeared in the literature recently as a convergence improvement trick. The novelty of the proposed methods lies in developing more sophisticated split/merge moves that fully harness the sub-clusters. In general the paper is well written but a few things needs to be clarified and investigated.

In particular:
* Augmenting your parameter space in general increases Monte Carlo error. As the authors state, convergence of the sub-clusters is necessary. How does this behaviour scale?
* Equation (20) presents an approximation of the proposal distribution. However, this is not sufficiently discussed. How could we safeguard against a bad approximation? Has the quality of the approximation been assessed? Similarly for Equation (24).
* lines 405-412: if none of the algorithms has converged, it makes little sense to compare which algorithm is ‘approaching’ a solution.
* Equations 6-10: there are two columns labelled “Priors” and “Posteriors”, but some of the prior distributions shown are conditional on the data.

Typos & minor comments:
* line 145 “the meaning of,”
* line 238 "the prior distributions"....
* line 253 "constructs"
* line 290 ","
* v hat bar looks confusing - how about replacing is by v’bar?
Summary: The idea in the paper is interesting and useful but more thorough investigation would be required.

Submitted by Assigned_Reviewer_20

This paper introduces a split merge MCMC algorithm for HDP topic modelling based on auxiliary "sub-clusters." One pair of sub-clusters represents a potential split in an existing topic, which are then used to construct proposals for split merge. The paper provides two types of proposals: a local and a global one. The local proposal draws topic assignments for a clusters' words according to the parameters of the associated subclusters, whereas the global move draws topic assignments for all words. The split merge moves are interleaved with a restricted Gibbs sampler, which can operate in parallel. The model is compared against two models, "direct assignment" (DA) of the original HDP paper, and "finite symmetric Dirichlet" (FSD) which approximates the DP with a high dimensional symmetric Dirichlet (and is also parallelizable). The proposed model is empirically shown to improve the convergence speed of the HDP based topic model over the DA and FSD.

The paper is well written and clear, though more discussion on the intuition of the split merge moves would be beneficial, particularly as this is the core contribution of the paper. Ideally there would be a comparison against a simpler baseline model, such as a DP topic model.

minor issues:
line 359: sufficient statistics -> summary statistics
Summary: The paper introduces a parallelizable split merge MCMC algorithm for the HDP, suitable for corpora as large as the NYTimes dataset. The inference method is compared against two other inference algorithms for the HDP, demonstrating better convergence properties.

Submitted by Assigned_Reviewer_31

Summary: The paper develops a MCMC algorithm for the Hierarchical Dirichlet process (HDP) that accommodates splits and merges and can be parallelized. Issues with respect to developing effective splits are addressed and controlled experiments demonstrating the algorithm's effectiveness are presented.

This is a well written, technically sound paper. The primary contribution lies in extending the DP sub cluster framework of [1] to HDPs. Although, this is somewhat incremental, the innovations required to transition the ideas in [1] to HDP inference are significant enough to warrant publication.

Detailed Comments:

1) Global split/merge moves: I like these moves and it is certainly believable that to escape certain "local optima" moves that distribute a topic's mass globally are required. However, given the significant extra cost of having to loop through all data instances it is important to clearly demonstrate the benefit of incorporating such moves. To this end, the experiments with and without the global moves on the toy bars dataset are not convincing. The authors do mention that they notice significant improvements on the larger real world datasets. I would like to see an additional plot added to Figures 4 and 5 from a handicapped version of the algorithm that uses only local moves.

2) It appears that at each iteration, the topic to be globally split is being chosen uniformly at random. Perhaps a more data driven proposal could be utilized here. For example, the negative cross entropy measure (C) defined in lines 375-376 provides a nice heuristic for determining "confused" topics. A topic, many of whose words have high probability under other topics, could be selected with higher probability for a split.

3) The issue of approximating the Hastings ratio for the merge moves is somewhat confusing. Are the MCMC guarantees maintained in spite of the approximation?

4) Is there a reason no experimental comparison against Wang and Blei's SAMS sampler is presented? The code is publicly available and should really be compared against.

Summary: Overall, this is an interesting paper that takes another step towards scaling up Bayesian nonparametric models. There are some minor concerns that the authors need to address (see comments above).
Author Feedback
Author rebuttal: We thank the reviewers for their useful comments. Clarifications are below and will be included in any final submission.

R13 and R31 inquired about the approximation used in Eq 20. We agree that a more rigorous analysis would help. However we do not believe that this is possible and provide the following intuition. The sub-clusters are essentially the same as the regular-clusters, except that there are only 2 per cluster. As such, we expect sub-cluster parameters to be similar to regular-cluster parameters. In the stationary distribution, the inferred sub-clusters after a merge will also be similar to the original clusters that merged. This approximation is used in Eq 20 instead of running an embedded MCMC scheme to infer the proposed sub-clusters.

===R13===
Augmenting your parameter space in general increases Monte Carlo error. As the authors state, convergence of the sub-clusters is necessary. How does this behaviour scale?
We have empirically found that the additional sub-clusters converge very quickly within only a few iterations, even for large datasets. Furthermore, since the auxiliary variables are modeled as being generated from the original HDP model, they do not add complexity to the original sample space.

lines 405-412: if none of the algorithms has converged, it makes little sense to compare which algorithm is ‘approaching’ a solution.
We agree. However, by using two different initializations (1 or 50 initializations), we can see that the sample paths for the proposed algorithm are beginning to converge, whereas the other algorithms are still quite far apart.

Equations 6-10: there are two columns labelled “Priors” and “Posteriors”, but some of the prior distributions shown are conditional on the data.
This discrepancy results from the auxiliary variables being generated from the data. We will change “Prior Distributions” to “Generative Distributions” to avoid confusion.

Changing v hat bar to v’bar?
The notation is a bit overwhelming. However, throughout the paper, hats represent proposals, and bars represents auxiliary variables. Changing to v' does not follow the convention.

===R20===
Re: intuition of the split merge moves
Split merge moves have recently been exploited [7,11-14] to greatly improve convergence. Intuition of the HDP-SubCluster splits are given in Lines 159-161 and 300-303.

Re: comparison to baseline model (e.g. DP topic model)
Comparisons with a DP topic model analyze the quality of the model in fitting the data. Our goal is not to show whether HDP models are adequate for topic modeling. Rather, we show that our *inference* procedure in HDPs improves over previous methods on the *same* model.

===R31===
1) Global split/merge moves: ... I would like to see an additional plot added to Figures 4 and 5 from a handicapped version of the algorithm that uses only local moves.
Thank you for the great suggestion! They were omitted because Figure 4 is already cluttered, but we agree that it would make a useful point. When global splits are not used, the HOW likelihood and the number of clusters plateau much earlier. However, because space is limited, we may have to put this plot in the supplemental.

2) It appears that at each iteration, the topic to be globally split is being chosen uniformly at random. Perhaps a more data driven proposal could be utilized here...
This is also a great suggestion. We believe such an approach could further improve convergence. However, this complicates the narrative and adds another term to the Hastings ratio. Furthermore, the reverse move which selects the same topics may not be as easy to compute. Regardless, it is definitely something to consider.

4) Is there a reason no experimental comparison against Wang and Blei's SAMS sampler is presented?
The code is publicly available, but it would take a considerable effort to get the HOW likelihood vs. time plots we show. Additionally, accurate comparisons are impossible since the code base is completely different. We can conclude from their results (Fig 7 in [7]) that the HDP-SAMS often decreases performance. Previous literature [1] has already shown that DP-SAMS performs worse than DP-SubClusters. Furthermore, non-parallelizable split/merge algorithms such as SAMS are not scalable to the dataset sizes we considered (Line 407). We believe that these are convincing enough to not warrant a comparison.